# Association between antenatal diagnosis of late fetal growth restriction and educational outcomes in mid-childhood: A UK prospective cohort study with long-term data linkage study

**Laurentya Olga** [1], **Ulla Sovio** [1], **Hilary Wong** [2], **Gordon Smith** [1], **Catherine Aiken** [1]*

**1** Department of Obstetrics and Gynaecology and NIHR Cambridge Biomedical Research Centre, University of Cambridge, Cambridge, United Kingdom, **2** Department of Paediatrics, University of Cambridge, Cambridge, United Kingdom

* cema2@cam.ac.uk

**Data Availability Statement:** The raw data from the POPS study are not publicly available, due to the nature of the consent given by participants in

## Abstract

### Background

Fetal growth restriction (FGR) is associated with a suboptimal intrauterine environment, which may adversely impact fetal neurodevelopment. However, analysing neurodevelopmental outcomes by observed birthweight fails to differentiate between true FGR and constitutionally small infants and cannot account for iatrogenic intervention. This study aimed to determine the relationship between antenatal FGR and mid-childhood (age 5 to 7 years) educational outcomes.

### Methods and findings

The Pregnancy Outcome Prediction Study (2008–2012) was a prospective birth cohort conducted in a single maternity hospital in Cambridge, United Kingdom. Clinicians were blinded to the antenatal diagnosis of FGR. FGR was defined as estimated fetal weight (EFW) <10th percentile at approximately 36 weeks of gestation, plus one or more indicators of placental dysfunction, including ultrasonic markers and maternal serum levels of placental biomarkers. A total of 2,754 children delivered at term were divided into 4 groups: FGR, appropriate-for-gestational age (AGA) with markers of placental dysfunction, healthy small-for-gestational age (SGA), and healthy AGA (referent). Educational outcomes (assessed at 5 to 7 years using UK national standards) were assessed with respect to FGR status using regression models adjusted for relevant covariates, including maternal, pregnancy, and socioeconomic factors.

Compared to healthy AGA (*N* = 1,429), children with FGR (*N* = 250) were at higher risk of "below national standard" educational performance at 6 years (18% versus 11%; aOR 1.68; 95% CI 1.12 to 2.48, *p* = 0.01). By age 7, children with FGR were more likely to perform below standard in reading (21% versus 15%; aOR 1.46; 95% CI 0.99 to 2.13, *p* = 0.05),

the original study. Requests for POPS supplemental information, including raw data, can be made to Mrs Sheree Green-Molloy at the Department of Obstetrics and Gynaecology, Cambridge University, UK (paoandghod@medschl.cam.ac.uk). The educational data used in the study are a bespoke extract from the National Pupil Database (NPD) containing anonymised individual pupil level results of routinely administered national assessments. The data controller for the NPD is the UK Department for Education, Department of Work and Pensions, Higher Education Statistics Authority, HM Revenue & Customs. The NPD was accessed for this study via the Office of National Statistics Secure Research Service (ONS SRS) under a data-sharing agreement with the University of Cambridge. Access to NPD is available to researchers on application: https://www.gov.uk/guidance/apply-for-department-for-education-dfe-personal-data (last accessed 17th Feb 2023).

**Funding:** This work was supported by an Action Medical Research grant (GN2788 to CEA). The POP study was funded by the NIHR Cambridge Biomedical Research Centre (to GS) and supported by the NIHR Cambridge Clinical Research Facility. The funders had no role in study design, data collection and analysis, decision to publish, or preparation of the manuscript.

**Competing interests:** I have read the journal's policy and the authors of this manuscript have the following competing interests: GS has no direct conflict of interest. GS has received research support from Roche Diagnostics Ltd, Illumina and Sera Prognostics (fetal growth restriction, preeclampsia and preterm birth). GS's department has received payment from Roche for a talk given by GS (fetal growth restriction). GS has been a paid consultant to GSK (preterm birth) and is a member of a Data Monitoring Committee for GSK trials of RSV vaccination in pregnancy. GS and US are two of three named inventors on a patent application (PCT/GB2020/053312) filed by Cambridge Enterprise for novel predictive test for fetal growth disorder. GS serves on PLOS Medicine's editorial board. The other authors have no conflicts of interest including financial interests, activities, relationships, and affiliations to declare.

**Abbreviations:** AC, abdominal circumference; AFP, alpha fetoprotein; AGA, appropriate-for-gestational age; aOR, adjusted odds ratio; CI, confidence interval; EFW, estimated fetal weight; FGR, fetal growth restriction; HES, Hospital Episode Statistics; IMD, index of multiple deprivation; LGA, large-for-gestational age; MoM, multiples of the median for gestational age; PAPP-A, pregnancy-associated plasma protein-A; POPS, Pregnancy

writing (28% versus 23%; aOR 1.46; 95% CI 1.02 to 2.07, $p = 0.04$), and mathematics (24% versus 16%; aOR 1.49; 95% CI 1.02 to 2.15, $p = 0.03$). This was consistent whether FGR was defined by ultrasound or biochemical markers. The educational attainment of healthy SGA children ($N = 126$) was comparable to healthy AGA, although this comparison may be underpowered. Our study design relied on linkage of routinely collected educational data according to nationally standardised metrics; this design allowed a high percentage of eligible participants to be included in the analysis (75%) but excludes those children educated outside of government-funded schools in the UK. Our focus on pragmatic and validated measures of educational attainment does not exclude more subtle effects of the intrauterine environment on specific aspects of neurodevelopment.

## Conclusions

Compared to children with normal fetal growth and no markers of placental dysfunction, FGR is associated with poorer educational attainment in mid-childhood.

## Author summary

### Why was this study done?

- Previous studies have reported poor neurodevelopmental outcomes as a long-term consequence of low birth weight.

- However, analysis on the basis of birth weight alone does not distinguish fetal growth restriction (FGR) from a healthy small phenotype.

### What did the researchers do and find?

- Prospectively collected and meticulously phenotyped antenatal data from 2,754 children was linked to nationally validated measures of educational outcomes in mid-childhood.

- Attending clinicians were blinded to the antenatal diagnosis of FGR to avoid bias in clinical management, e.g., iatrogenic early-term delivery.

- We found that children with antenatally diagnosed FGR had approximately 70% and approximately 50% greater risk of not attaining educational standards at ages 6 and 7 (in reading, writing, and mathematics), respectively.

### What do these findings mean?

- FGR is associated with poorer educational in mid-childhood, and this association cannot be explained by iatrogenic harm arising from earlier delivery at term.

- This finding has important implications for the clinical management of FGR diagnosed in late pregnancy.

Outcome Prediction Study; sFlt1:PlGF, soluble fms-like tyrosine kinase 1:placental growth factor ratio; SGA, small-for-gestational age; UMB-PI, umbilical artery pulsatility index; UT-PI, uterine artery pulsatility index.

## Introduction

Fetal growth restriction (FGR) is defined as the failure to achieve genetically determined growth potential in utero. FGR is a major cause of morbidity and mortality during the perinatal period but is also associated with lifelong adverse consequences [1]. A major long-term concern is the potential for suboptimal intrauterine environments to lead to impaired fetal neurodevelopment, which could explain previously described associations between low birth weight and poorer educational outcomes in childhood [2]. However, studying the relationship between FGR and long-term outcomes is complicated for several reasons. First, babies that are born small-for-gestational-age (SGA) include both true cases of FGR and those who are healthy but constitutionally small. Hence, analysing associations by observed birth weight alone may fail to identify the true impact of FGR. Identifying true cases of FGR requires combining ultrasonic estimates of fetal size with other antenatal indicators of placental dysfunction [1]. Second, antenatal diagnosis of suspected FGR leads to enhanced fetal monitoring and intervention, principally, early-term delivery (37 to 38 weeks gestational age). Intervention could improve outcomes by reducing the exposure of the fetus to a hostile intrauterine environment. However, early-term delivery is itself associated with poorer educational outcome [2]. Hence, it is also possible that poor long-term educational outcome in SGA babies is caused by intervention, as recently proposed in a study of Australian children [3]. There is currently a critical gap in knowledge about the natural history of antenatally diagnosed FGR and long-term neurodevelopmental outcome. Addressing this gap requires a prospective study with detailed phenotyping of intrauterine growth, where clinical management was not influenced by study data and robust long-term data on educational attainment is available.

The aim of this study was to determine whether antenatally diagnosed FGR predicted poorer educational attainment in childhood by employing data linkage between a large prospective UK birth cohort and a national registry of educational data.

## Methods

### Study design and data sources

The Pregnancy Outcome Prediction Study (POPS) recruited unselected nulliparous women with singleton pregnancies at The Rosie Hospital, Cambridge, England, between January 2008 and July 2012 ($n = 4,212$). The full cohort design and demographics have previously been described [4–6]. Women were recruited following confirmation of a single viable fetus and dating of the pregnancy based on the crown-rump length measured by ultrasound (<14 weeks of gestation). Three further research visits were conducted at 20, 28, and 36 weeks of gestation where fetal biometry, liquor, and pulsatility indices in the uterine and umbilical arteries were measured using Doppler flow velocimetry. Maternal blood samples were obtained at all research visits. Assays for pregnancy-associated plasma protein-A (PAPP-A), alpha fetoprotein (AFP), and soluble fms-like tyrosine kinase 1:placental growth factor ratio (sFlt1:PlGF) were performed on a Roche Diagnostics Cobas e411 platform, as previously described [4,6–9]. Antenatal, delivery, and neonatal outcome data were collected from various sources, including questionnaires, paper records, and electronic medical record systems. The cohort has several features making it ideal to study the relationship between FGR and educational attainment. First, all participants were meticulously phenotyped with respect to markers of FGR. [4,9] Although there is no universally accepted definition of FGR, all the elements included in the current definition have been previously reported in the cohort [6–10], and the rationale for their inclusion is discussed elsewhere [1,11]. Second, all antenatal measurements were made prospectively within defined gestational time frames using standardised investigations,

allowing deep phenotyping of fetal growth [4,5]. Third, clinicians were blinded to the results of all research scans and investigations [6], reducing confounding of the relationship between antenatal factors and educational outcome by clinical interventions.

Educational outcomes were obtained from the National Pupil Database, a national record-level data resource curated by the UK Department for Education (DfE; last follow-up at the end of school year 2019). All fully/partially state-funded schools in England have a mandatory reporting requirement to return individual-level pupil data on an annual basis [12]. Standardised pupil assessment data were available for the cohort at ages 5, 6, and 7 (see S1 and S2 Appendix files for details of assessment). Binary outcomes (below versus at or above standard) were generated for attainment aged 5 and aged 6. At aged 7, children were assessed separately on 4 educational domains (reading, writing, mathematics, and science), each of which was expressed as a binary outcome [13]. Because participants started school over the course of 5 different school years, analyses were adjusted for year of starting school.

We obtained written informed consent for participation in the POPS cohort and collection of immediate outcome data. In order to obtain long-term outcome data, having confirmed the vital status of the mother and child, we contacted POPS participants in 2018 with information about the proposed linkage and allowing participants to opt out (45 exercised this choice). All data were linked anonymously and accessed via the Office of National Statistics Secure Research Service. Approvals were obtained from Cambridgeshire 2 Research Ethics Committee (antenatal study), Cambridge Central Research Ethics Committee, and the UK Confidentiality Advisory Board (linkage study).

Data cleaning was performed upon POPS data by cross-checking implausible clinical data values with paper records or, if unsolved, setting the values to missing. This process was rigorously checked and supervised to avoid unnecessary data exclusions.

The reporting of this study conforms to the REporting of studies Conducted using Observational Routinely-collected Data (RECORD) extension of the Strengthening the Reporting of Observational Studies in Epidemiology (STROBE) guidelines (S1 RECORD STROBE Extension Checklist).

## Outcomes, exposures, and covariates

The main outcome of the study was educational attainment assessed at 5 years, 6 years, and 7 years, represented, respectively, by a binary outcome at each age, with the assessment aged 7 divided into 4 key learning domains (S2 Appendix).

To avoid a confounding effect of prematurity, infants delivered prior to 37+0 weeks of gestation were excluded from analyses. The estimated fetal weight (EFW) was calculated using the Hadlock III formula [14] from individual biometry measurements (abdominal circumference, head circumference, femur length, biparietal diameter) obtained at antenatal ultrasound at approximately 36 weeks as previously described [6]. The calculated EFW was converted to a gestational age-adjusted percentile, and participants were classified as either appropriate-for-gestational age (AGA; EFW $\geq$10th and $\leq$90th centiles) or small-for-gestational age (SGA; EFW <10th centile) [15]. Infants born large-for-gestational age (LGA; EFW >90th) were excluded from analyses.

We divided the cohort into categories that were prespecified according to clinical understanding of the risk associated with poor fetal growth and placental dysfunction (S1 Appendix). Antenatally determined EFW categories were used to subdivide the cohort according to the presence or absence of any markers for FGR to give 4 groups: (i) FGR; (ii) healthy SGA; (iii) AGA with markers of placental dysfunction; and (iv) healthy AGA. Healthy AGA was used as the referent group throughout. FGR was defined as an EFW <10th centile combined

with one or more defined markers of placental dysfunction. Healthy SGA was defined as an EFW <10th in the absence of any features of placental dysfunction. Markers of placental dysfunction were defined as any of the following (S1 Appendix): (i) lowest decile of abdominal circumference (AC) growth velocity between 20 and 36 weeks; (ii) EFW <third centile at 36 weeks; (iii) umbilical artery pulsatility index (UMB-PI) in the highest decile at 36 weeks; (iv) uterine artery pulsatility index (UT-PI) in the highest decile at 20 weeks; (v) maternal PAPP-A <0.4 multiples of the median for gestational age (MoM) at 12 weeks [7]; (vi) maternal sFlt1: PlGF ratio >38 at 36 weeks [8]; and (vii) maternal AFP >2.0 MoM at 20 weeks [7].

Covariates included in the analysis were sex; school funding type (academy, community, or voluntary; standard models of UK state school funding; [16]); maternal factors, including BMI, partner status, age, smoking status during pregnancy, occupation, and ethnicity (all recorded at recruitment); index of multiple deprivation (IMD score: 2007; [17]), gestational age, presence of significant childhood morbidity, season of birth, and school year of assessment.

To minimise the impact of significant childhood morbidities that were not linked to intrauterine development on the analyses, we adjusted for relevant morbidities. Data were extracted from Hospital Episode Statistics (HES; provided via NHS Digital) relating to (i) emergency department attendances, (ii) hospital inpatient stays, and (iii) hospital outpatient appointments to compile a prespecified list of conditions (S1 Table).

## Statistical analyses

The study analyses followed a prespecified statistical analysis plan agreed by all study authors (S1 Appendix). Children with relevant childhood morbidities (S1 Table) were included in the analyses, and the models were instead adjusted for this factor, rather than excluding this small number of children as their exclusion did not materially alter the results. Other primary analyses were conducted as planned. Sensitivity analyses by using EFW threshold <20th percentile or excluding women with ultrasound scan >35 weeks of gestational age did not substantively alter the results and so these have not been reported here (S7 and S8 Tables).

Baseline demographics were described using numerical and categorical variables, presented as mean ± standard deviation and N (%), respectively. The associations between exposure groups and educational outcomes were modelled using multivariable logistic regression, adjusted for identified covariates.

Covariates that had a small proportion of missing values, including maternal BMI, partner status, and school funding status, were imputed using chained equations (MICE) under a "missing-at-random" assumption [18]. The R package "mice" was used to generate 20 imputed datasets, using linear regression for continuous variables and logistic regression for categorical variables. These variables included in the imputation models: gestational age, birth weight centiles, delivery modes, maternal factors (age, BMI, occupation, ethnicity, partner status, alcohol consumption, smoking status), IMD, school funding type, and early years z-scores. Analyses run on each dataset were pooled according to Rubin's rules [18], and imputed values were found comparable to observed values. No imputation was performed with respect to FGR markers as the missing percentages ranged from 0% to 1.4%. Only children with at least one "present" FGR marker or all "absent" FGR markers were included in the analyses.

We assumed prospectively that approximately 75% of the cohort would have linked data available and that approximately 10% of the cohort would have EFW <10th centile, with 80% being defined as AGA. Our power calculations were thus based on the assumption that approximately 6% to 7% might be expected to meet our definitions of FGR, with approximately 50% meeting the definition of healthy AGA. The study was thus powered to detect a 50% difference in the percentage of FGR children failing to attain expected educational

standards at age 5 versus healthy AGA at 80% power with alpha = 0.05. In the event, the data available were such that the FGR versus healthy AGA comparison was adequately powered at 50% difference between groups, but due to smaller numbers, the healthy SGA versus healthy AGA comparison was not powered at 50% difference.

All analyses were conducted using R version 3.6.1. [19]. Where p-values (two-sided) are reported, an alpha level of 0.05 was considered statistically significant.

## Results

Approximately 65% of all antenatally recruited participants (2,754/4,212) were included in the analytic sample, representing 75% of those eligible for data linkage (2,754/3,677) (Fig 1). Mean

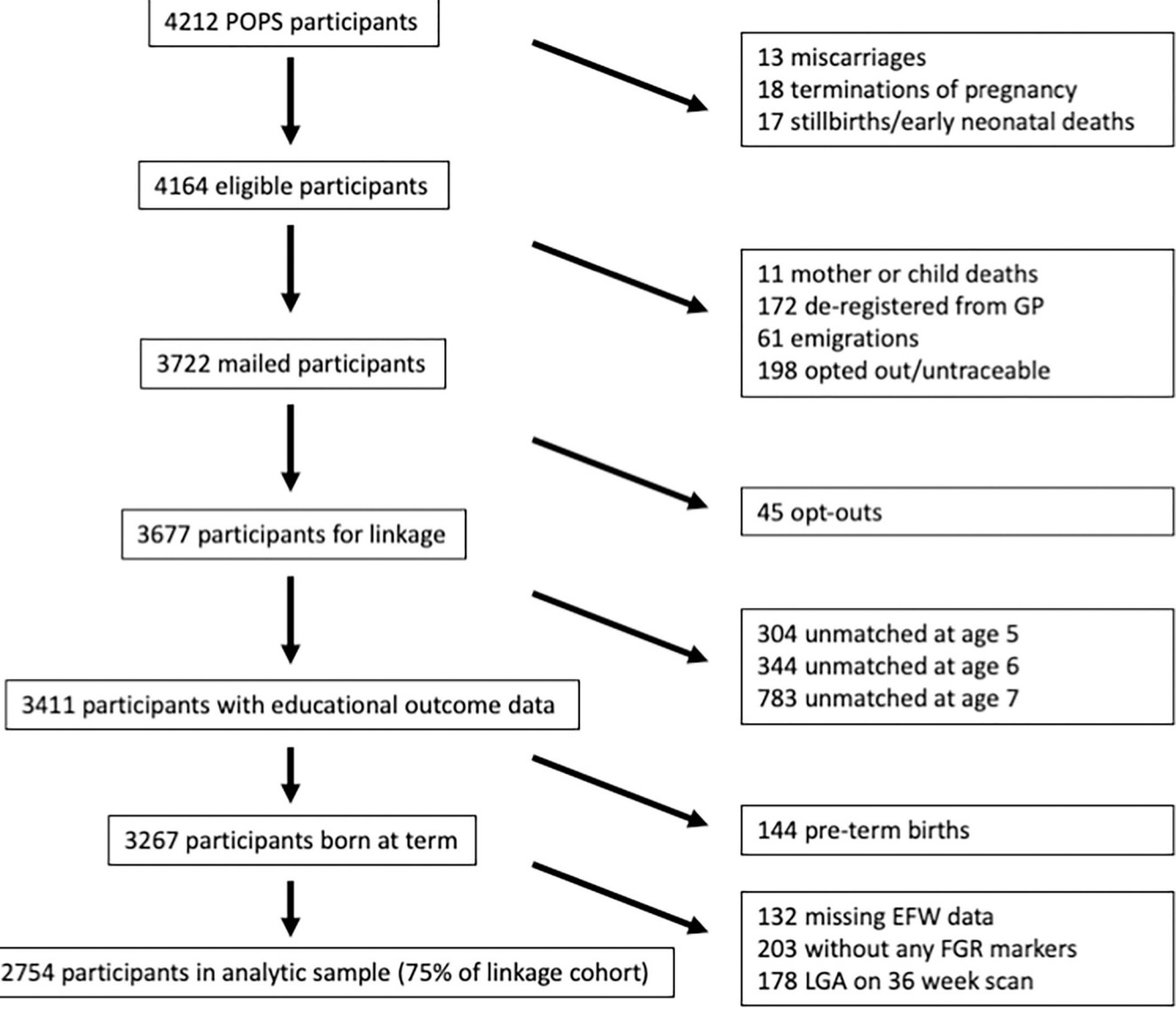

**Fig 1. Cohort profile.** Number of participants from recruitment to POP study through identification of analytic sample. Approximately 75% of participants who were eligible for linkage (2,754/3,677) are included in the analytic sample, which represents 65% of the total participants originally recruited (2,754/4,212). EFW, estimated fetal weight; FGR, fetal growth restriction; GP, General Practitioner; LGA, large-for-gestational age; POPS, Pregnancy Outcome Prediction Study.

**Table 1. Baseline characteristics of all eligible participants and the analytic sample.**

| Characteristic | Analytic sample (N = 2,754) | Excluded participants (N = 1,410) | All POPS participants (N = 4,164) |
|---|---|---|---|
| **Maternal characteristics** | | | |
| Age, yrs, mean (SD) | 29.8 (5.1) | 30.3 (5.1) | 29.9 (5.1) |
| Age stopped FTE, y, mean (SD) | 20.7 (3.7) | 21.6 (4.2) | 21 (3.9) |
| Missing, No. (%) | 79 (2.9) | 47 (3.3) | 126 (3) |
| Height, cm, mean (SD) | 165.1 (6.4) | 165.2 (6.6) | 165.2 (6.4) |
| BMI, kg/m², mean (SD) | 25.1 (4.7) | 25.2 (4.7) | 25.1 (4.7) |
| Missing, No. (%) | 1 (0.04) | 0 | 1 (0.02) |
| Ethnicity, No. (%) of White/European | 2,592 (94.1) | 1,269 (90) | 3,861 (92.7) |
| Missing, No. (%) | 37 (1.3) | 33 (2.3) | 70 (1.7) |
| Smoking history, No. (%) of never smoked | 1,596 (58) | 879 (62.3) | 2,475 (59.4) |
| Alcohol consumption, No. (%) of not drinking | 2,617 (95) | 1,359 (96.4) | 3,975 (95.5) |
| Missing, No. (%) | 1 (0.04) | 1 (0.1) | 2 (0.1) |
| Partner status, No. (%) of have partner | 2,647 (96.1) | 1,364 (96.7) | 4,011 (96.3) |
| Missing, No. (%) | 57 (2.1) | 3 (0.2) | 3 (0.1) |
| IMD, mean (SD) | 10.1 (6.6) | 10.6 (6.5) | 10.3 (6.5) |
| Missing, No. (%) | 0 | 61 (4.3) | 171 (4.1) |
| **Perinatal characteristics** | | | |
| Gestational age, wk, mean (SD) | 40.2 (1.2) | 39.5 (2.3) | 40 (1.7) |
| Missing, No. (%) | 3 (0.1) | 6 (0.4) | 9 (0.2) |
| Sex, No. (%) of female | 1,396 (50.7) | 669 (47.5) | 2,065 (49.6) |
| Missing, No. (%) | 0 | 0 | 1 (0.02) |
| Birth seasonality, No. (%) of children born in autumn | 756 (27.5) | 367 (26) | 1,123 (27) |
| Mode of delivery, No. (%) of vaginal delivery | 2,046 (74.3) | 962 (68.2) | 3,008 (72.2) |
| Missing, No. (%) | 4 (0.1) | 7 (0.5) | 11 (0.3) |
| Birth weight, centile, mean (SD) | 43.7 (25.1) | 49.38 (27.4) | 45.7 (26) |
| Missing, No. (%) | 0 | 1 (0.1) | 1 (0.02) |
| SGA at delivery, No. (%) | 252 (9.2) | 116 (8.2) | 368 (8.8) |
| Missing, No. (%) | 0 | 1 (0.1) | 1 (0.02) |

BMI, body mass index; FTE, full-time education; IMD, index of multiple deprivation; SGA, small-for-gestational age; wk, weeks; yrs, years.

gestational age at delivery was 40 weeks, and the average birthweight was approximately 43rd centile (Table 1). The demographic characteristics of the analytic sample reflected the total eligible study participants, and there were no material differences (Tables 1 and S2).

The analytic sample was distributed as follows: FGR 9% (250/2,754), AGA with markers of placental dysfunction 34.5% (949/2,754), healthy SGA 4.6% (126/2,754), and healthy AGA 51.9% (1,429/2,754) (S3 Table).

## Educational attainment aged 5 and 6

Of participants with linked educational outcome data at age 5 and 6, 20% (559/2,735) and 12% (314/2,699), respectively, did not meet expected educational standards. Children with antenatal FGR had a higher chance of not achieving these standards at age 5 compared to those who were healthy AGA (25% versus 19%; adjusted odds ratio [aOR] 1.33; 95% CI 0.93 to 1.89 p = 0.11; Table 2) and at age 6 (18% versus 11%; aOR 1.68; 95% CI 1.12 to 2.48, p = 0.01; Table 2).

**Table 2. Association between FGR and educational attainment aged 5–7 years.**

| Assessment | FGR Frequency (%) | Healthy AGA Frequency (%) | Unadjusted OR (95% CI) | p | Adjusted OR (95% CI) | p |
|---|---|---|---|---|---|---|
| Age 5 | 63/250 (25) | 194/942 (21) | 1.41 (1.02–1.92) | **0.03** | 1.33 (0.93–1.89) | 0.11 |
| Age 6 | 45/246 (18) | 100/929 (11) | 1.84 (1.26–2.63) | **0.001** | 1.68 (1.12–2.48) | **0.01** |
| Age 7—Reading | 47/223 (21) | 122/801 (15) | 1.46 (1.01–2.07) | **0.04** | 1.46 (0.99–2.13) | **0.05** |
| Age 7—Writing | 62/223 (28) | 186/802 (23) | 1.44 (1.03–1.98) | **0.03** | 1.46 (1.01–2.07) | **0.04** |
| Age 7—Mathematics | 53/223 (24) | 130/802 (16) | 1.62 (1.14–2.28) | **0.006** | 1.49 (1.02–2.15) | **0.03** |
| Age 7—Science | 23/223 (10) | 73/802 (9) | 1.03 (0.63–1.62) | 0.9 | 0.98 (0.58–1.58) | 0.92 |

AGA, appropriate-for-gestational age; CI, confidence interval; FGR, fetal growth restriction; OR, odds ratio.

No/total (%) of participants of FGR and healthy AGA who failed to achieve the expected educational standards aged 5–7 years are displayed. Unadjusted and adjusted ORs with 95% CIs of FGR are displayed with antenatal healthy AGA as the referent group. Significant p-values are typed in **bold.**

### Educational attainment aged 7

At age 7, the proportions failing to attain expected educations standards were 16% for reading (N = 381/2,354), 22% for writing (N = 525/2,351), 17% for mathematics (N = 396/2,351), and 10% for science (N = 226/2,351). Children with an antenatal diagnosis of FGR had an increased risk of not achieving the expected standard in reading (21% versus 15%; aOR 1.46; 95% CI 0.99 to 2.13, p = 0.05; Table 2), writing (28% versus 23%; aOR 1.46; 95% CI 1.02 to 2.07, p = 0.04; Table 2), and mathematics (24% versus 16%; aOR 1.49; 95% CI 1.02 to 2.15, p = 0.03; Table 2) compared to the referent group, but there was no difference between the groups in the science domain.

There were no significant differences in educational attainment at any age when children who were either healthy SGA or AGA with placental dysfunction were compared to those who were healthy AGA (Tables A and B in S4 Table). However, the 95% CI surrounding the point estimates of effect for the healthy SGA group were wide, reflecting smaller group size and hence greater levels of uncertainty about where the true value may lie. The associations were very similar when cases of FGR defined using ultrasonic markers were compared to cases defined by biochemical markers of placental dysfunction (Tables A-F in S5 Table). FGR was also associated with mid-childhood educational performance in unadjusted models (Tables A-F in S5 Table).

### Educational attainment stratified by birthweight

Participants with FGR were stratified by actual birthweight (<10th centile versus ≥10th centile). The association between antenatal FGR and later educational performance remained only in those in whom FGR was correctly identified (actual birthweight <10th centile; S6 Table). Due to the smaller group sizes, the association no longer met the prespecified threshold for statistical significance in all analyses, but the direction of the association was consistent.

### Discussion

Fetuses with evidence of antenatal FGR had increased likelihood of lower educational attainment through mid-childhood. The association was consistent across serial assessments aged 5 through 7. Although not statistically significant at age 5, potentially due to the nature of the assessment or the higher percentage of children overall who do not meet expected standards at this stage, the estimate of effect was broadly similar across all ages studied, suggesting that the impact of FGR on educational performance is consistent across the early school years. By age

7, a clear difference could be detected between educational domains, with children who exhibited FGR being more likely than other children to perform poorly in reading, writing, and mathematics. Our findings suggest that a suboptimal intrauterine environment caused by placental insufficiency (e.g., reduced partial pressure of oxygen, reduced supply of nutrients, or chronic pro-inflammatory response [20,21]) is associated with poorer future educational attainment. Fetal neural development relies on a tightly regulated and highly stereotyped series of transcriptional and epigenetic modulations [22] that may be vulnerable to environmental disruption induced by placental dysfunction, for example, in the context of preeclampsia [23]. Our results are in keeping with previous findings suggesting lower cognitive, linguistic, and communication skills [24,25], and also lower social adjustment and emotional readiness for school [26] in children affected by FGR.

A key strength of the present study is that the imaging and biochemical data used to define FGR were not revealed to the attending clinicians, an important feature of the prospective study design to avoid multiple biases. FGR detected near term would usually lead to early-term delivery and the influence of this on the outcome is complex, incompletely understood, and is likely to differ between true cases of FGR and healthy constitutionally small infants. Early-term delivery could potentially mask associations by abbreviating the duration of exposure to the hostile intrauterine environment in true cases of FGR. Alternatively, birth at early term is associated with an increased risk of special educational needs, which is part of the continuum of risk associated with earlier delivery across the whole range of gestational age [2]. Hence, it is possible that early-term delivery, indicated by the antenatal suspicion of FGR, could actually be an iatrogenic cause of poor educational attainment, particularly in constitutionally small fetuses. In the present study, these biases were avoided by prospective data collection and blinding of the results to the attending clinicians.

The current analysis is highly relevant for clinicians considering intervention for suspected FGR near term. Early-term delivery is currently considered in this context to reduce the risk of stillbirth, as FGR infants have an 8-fold risk of antepartum intrauterine fetal death prior to the onset of labor, as well as increased risks of intrapartum and neonatal death [27]. If the association between FGR and poor educational outcomes was secondary to earlier delivery at term rather than to the disease process of FGR, as has been proposed [3], clinicians may consider nonintervention, with a subsequent increased risk of stillbirth. Our data indicate that late FGR is an independent predictor of poor educational outcomes and that this association cannot be explained by iatrogenic harm arising from earlier delivery at term. Consistent with this, a randomised controlled trial has previously shown that routine induction of labor at 37 weeks for EFW <10th percentile (without taking into account markers of FGR) was not associated with an overall increase or decrease in the risk of poor neurodevelopmental outcome in childhood [28]. There was no negative effect of early-term delivery despite the fact that many infants in the trial would have been constitutionally small. The neutral overall effect of early-term delivery in this trial may be explained by a balancing of benefit gained by early-term delivery in true cases of FGR versus harm to healthy constitutionally small infants. Hence, it is plausible that if early-term delivery is targeted only to true cases of FGR, neurodevelopmental outcomes might be improved by reducing fetal exposure to a suboptimal intrauterine environment. Moreover, better methods for the diagnosis of FGR might avoid iatrogenic harm by avoiding unnecessary intervention for constitutionally small fetuses. However, FGR remains a theoretical concept with no gold diagnostic standard. Our phenotyping in this study is derived from current best understanding and consensus regarding optimal markers [11,29]. Our observations provide a rationale for developing enhanced methods of screening and diagnosis for late FGR. Deep phenotyping of small fetuses may allow stratified antenatal care, targeting the intervention of early-term delivery specifically to FGR cases who are most likely to benefit. Hence,

FGR should be a priority area for discovery science methods to identify new predictors [11] and for interventional trials to determine the short- and long-term consequences of interventions [1].

### Limitations

This study has several limitations. First, it relies on educational data collected by the UK Department of Education. Although this is an extremely well-curated source with mandatory reporting requirements [12], only pupils in state-funded schools in England are included. It is therefore possible that pupils of higher socioeconomic status (who are more likely to attend private schools) and those of non-UK backgrounds (who may be more likely to attend school outside England) may be disproportionately missing. However, the high linkage rate and lack of significant demographic difference between the whole cohort and the analytic sample (Table 1) suggests that the study group was representative of the whole cohort.

Second, the cohort originates from a single centre in Cambridge, UK, and is less socioeconomically and ethnically diverse than the whole population of the UK [4]. It was recruited in a relatively affluent area with a good state-funded education system [30]. This may reduce the general applicability of the results. However, the relative homogeneity of the population reduces the potential for noise and bias. Other novel findings from the study have previously been externally validated in demographically dissimilar populations [10,31,32].

Third, although adjustment for a wide range of potentially confounding variables was made, there are inevitably further unmeasured confounders that may influence childhood neurocognitive development [33,34].

Finally, since FGR lacks a gold standard universally accepted definition, there might be a possibility of misidentifying FGR or healthy AGA. However, all participants were meticulously phenotyped with respect to reported markers of FGR, as reported elsewhere [4,9].

### Conclusions

Compared with children with normal fetal growth and no markers of placental dysfunction, FGR is associated with poorer educational attainment in mid-childhood.

Data are obtained from the original dataset before imputation. For fields where there is no "missing" row, data were 100% complete. Maternal age was defined as age at recruitment. Maternal BMI was derived from weight measured at recruitment divided by the square of height ($kg/m^2$). All other maternal characteristics were either self-reported at the 20-week gestational age visit from examination of the clinical record or linkage to the hospital's electronic databases. Deprivation was quantified using IMD 2007 based on census data from the area of the mother's postcode [17]. Birth weight percentiles and z scores were calculated using UK 1990 growth reference [35]. SGA at delivery is defined as birth weight <10th percentile according to UK 1990 growth reference [35]. *$p < 0.05$ in comparisons of the analytic sample versus all eligible POPS participants.

Markers of placental dysfunction are defined as one or more of the following: low AC growth between 20 and 36 weeks, high uterine artery pulsatility index at 20 weeks, high umbilical artery pulsatility index at 36 weeks, EFW <third centile, low PAPPA, sFlt-1:PlGF ratio, high AFP.

For adjusted models, covariates included in all models: maternal factors (age at pregnancy, BMI at recruitment, ethnicity, occupation, partner status, smoking history), infant factors (gestational age, sex, birth seasonality, childhood physical health), socioeconomic factors (IMD, school funding type, academic year).

## Supporting information

**S1 Appendix. Data analysis plan.** Analysis plan for assessing whether prenatal diagnosis of late fetal growth restriction is predictive of later educational achievement.
(DOCX)

**S2 Appendix. Assessment of educational attainment aged 5, 6, and 7 years in the UK.**
(DOCX)

**S1 RECORD STROBE Extension Checklist. The RECORD statement—Checklist of items, extended from the STROBE statement, which should be reported in observational studies using routinely collected health data.** Reference: Benchimol EI, Smeeth L, Guttmann A, Harron K, Moher D, Petersen I, Sørensen HT, von Elm E, Langan SM, the RECORD Working Committee. The REporting of studies Conducted using Observational Routinely-collected health Data (RECORD) Statement. *PLoS Medicine* 2015; in press. *Checklist is protected under Creative Commons Attribution (CC BY) license. NA, not applicable.
(DOCX)

**S1 Table. Childhood medical conditions for model adjustment.** These conditions are not linked to intrauterine development but could impact childhood educational performance and therefore were adjusted in primary analyses. This prespecified morbidity list was defined in consultation with a paediatric consultant (HW). A full year of hospital episode statistics (HES) was obtained for each child, as it is highly likely that any child with a significant excludable health condition would have at least one HES-recorded appointment within a year. Children without any HES data recorded during the time frame (1 year) are assumed to not have any of the prespecified morbidities. There may be a small number of children who were being managed entirely via the private healthcare system or not have required any hospital management at all over the course of a year; however, given the medical complexity of the prespecified conditions, this is unlikely and would only apply to a very small number of children.
(DOCX)

**S2 Table. Ultrasonic biometry and markers of placental dysfunction of all POPS participants and the analytic sample.** Abbreviations: AC, abdominal circumference; AFP, alpha-feto protein; EFW, estimated fetal weight; PAPP-A, pregnancy-associated plasma protein-A; sFlt1: PlGF, soluble fms-like tyrosine kinase 1:placental growth factor ratio; UMB-PI, umbilical artery pulsatility index; UT-PI, uterine artery pulsatility index.
(DOCX)

**S3 Table. Baseline characteristics among exposure groups.** Values are median (IQR) or N (%) as appropriate. Maternal age was defined as age at recruitment. Maternal BMI was derived from weight measured at recruitment divided by the square of height ($kg/m^2$). All other maternal characteristics were either self-reported at the 20-week gestational age visit, from examination of the clinical record, or linkage to the hospital's electronic databases. Deprivation was quantified using the IMD 2007 based on census data from the area of the mother's postcode. Birth weight percentiles and z scores were calculated using UK 1990 growth reference. Abbreviations: AGA, appropriate-for-gestational age; BMI, body mass index; FGR, fetal growth restriction; GA, gestational age; IMD, index of multiple deprivation; IQR, interquartile range; SGA, small-for-gestational age.
(DOCX)

**S4 Table. Educational attainment aged 5–7 years by fetal growth status. Table A.** Rate of passing educational standard aged 5–7 years in all exposure groups. No/total (%) of

participants of each group who failed corresponding educational assessment are displayed. *$p < 0.05$ and **$p < 0.01$ versus healthy AGA (referent) based on chi-squared test. Abbreviations: AGA, appropriate-for-gestational age; FGR, fetal growth restriction; GA, gestational age. **Table B.** Association between educational attainment aged 5–7 years and presence of any markers of placental dysfunction by fetal growth status. Outcome: Not achieving expected educational standard at each age/domain (as appropriate). Odds ratios (OR; for unadjusted models) or adjusted odds ratios (aOR; for adjusted models) with 95% confidence intervals are displayed with antenatal healthy AGA (Total $N = 1,429$) as the referent group. Markers of placental dysfunction are defined as one or more of the following: low AC growth between 20–36 weeks, high uterine artery pulsatility index at 20 weeks, high umbilical artery pulsatility index at 36 weeks, EFW <third centile, low PAPPA, sFlt-1:PlGF ratio, and high AFP. For adjusted models, covariates included in all models: maternal factors (age at pregnancy, BMI at recruitment, ethnicity, occupation, partner status, smoking history), infant factors (gestational age, sex, birth seasonality, childhood physical health), socioeconomic factors (IMD, school funding type, academic year). Abbreviations: AC, abdominal circumference; AFP, alpha-feto protein; AGA, appropriate-for-gestational age; aOR, adjusted odds ratio; CI, confidence interval; EFW, estimated fetal weight; FGR, fetal growth restriction; OR, odds ratio; PAPP-A, pregnancy-associated plasma protein-A; sFlt1:PlGF, soluble fms-like tyrosine kinase 1:placental growth factor ratio; SGA, small-for-gestational age; UMB-PI, umbilical artery pulsatility index; UT-PI, uterine artery pulsatility index.
(DOCX)

**S5 Table. Associations between educational attainment aged 5–7 and markers of placental dysfunction by fetal growth status. Table A.** Association between educational attainment aged 5 and markers of placental dysfunction by fetal growth status. Outcome: Not achieving expected educational standard aged 5. Odds ratios (OR) with 95% confidence intervals are displayed with antenatal healthy AGA ($N = 1,418$) as the referent group. [a]Low AC velocity, High UT-PI, High UMB-PI, EFW <third centile, Low PAPP-A, sFlt-1:PlGF >38, High AFP; [b]Low AC velocity, High UT-PI, High UMB-PI, EFW <third centile; [c]Low PAPP-A, sFlt-1:PlGF >38, High AFP. Covariates included in fully adjusted models: maternal factors (age at pregnancy, BMI at recruitment, ethnicity, occupation, partner status, smoking history), infant factors (gestational age, sex, birth seasonality, childhood physical health), socioeconomic factors (IMD, school funding type, academic year). Abbreviations: AC, abdominal circumference; AFP, alpha-feto protein; AGA, appropriate-for-gestational-age; aOR, adjusted odds ratio; CI, confidence interval; EFW, estimated fetal weight; FGR, fetal growth restriction; NA, not applicable; PAPP-A, pregnancy-associated plasma protein-A; sFlt1:PlGF, soluble fms-like tyrosine kinase 1:placental growth factor ratio; SGA, small-for-gestational-age; UMB-PI, umbilical artery pulsatility index; UT-PI, uterine artery pulsatility index. **Table B.** Association between educational attainment aged 6 and markers of placental dysfunction by fetal growth status. Outcome: Not achieving expected educational standard aged 6. Odds ratios (OR) with 95% confidence intervals are displayed with antenatal healthy AGA ($N = 1,399$) as the referent group. [a]Low AC velocity, High UT-PI, High UMB-PI, EFW <third centile, Low PAPP-A, sFlt-1:PlGF >38, High AFP; [b]Low AC velocity, High UT-PI, High UMB-PI, EFW <third centile; [c]Low PAPP-A, sFlt-1:PlGF >38, High AFP. Covariates included in fully adjusted models: maternal factors (age at pregnancy, BMI at recruitment, ethnicity, occupation, partner status, smoking history), infant factors (gestational age, sex, birth seasonality, childhood physical health), socioeconomic factors (IMD, school funding type, academic year). Abbreviations: AC, abdominal circumference; AFP, alpha-feto protein; AGA, appropriate-for-gestational age; aOR, adjusted odds ratio; CI, confidence interval; EFW, estimated fetal weight; FGR, fetal

growth restriction; NA, not applicable; PAPP-A, pregnancy-associated plasma protein-A; sFlt1:PlGF, soluble fms-like tyrosine kinase 1:placental growth factor ratio; SGA, small-for-gestational age; UMB-PI, umbilical artery pulsatility index; UT-PI, uterine artery pulsatility index. Table C. Association between educational attainment aged 7 (Reading domain) and markers of placental dysfunction by fetal growth status. Outcome: Not achieving expected educational standard aged 7 in Reading domain. Odds ratios (OR) with 95% confidence intervals are displayed with antenatal healthy AGA ($N$ = 1,214) as the referent group. [a]Low AC velocity, High UT-PI, High UMB-PI, EFW <third centile, Low PAPP-A, sFlt-1:PlGF >38, High AFP; [b]Low AC velocity, High UT-PI, High UMB-PI, EFW <third centile; [c]Low PAPP-A, sFlt-1:PlGF >38, High AFP. Covariates included in all models: maternal factors (age at pregnancy, BMI at recruitment, ethnicity, occupation, partner status, smoking history), infant factors (GA, sex, birth seasonality, childhood physical health), socioeconomic factors (IMD, school funding type, academic year). Abbreviations: AC, abdominal circumference; AFP, alpha-feto protein; AGA, appropriate-for-gestational age; aOR, adjusted odds ratio; CI, confidence interval; EFW, estimated fetal weight; FGR, fetal growth restriction; NA, not applicable; PAPP-A, pregnancy-associated plasma protein-A; sFlt1:PlGF, soluble fms-like tyrosine kinase 1:placental growth factor ratio; SGA, small-for-gestational age; UMB-PI, umbilical artery pulsatility index; UT-PI, uterine artery pulsatility index. Table D. Association between educational attainment aged 7 (Writing domain) and markers of placental dysfunction by fetal growth status. Outcome: Not achieving expected educational standard aged 7 in Writing domain. Odds ratios (OR) with 95% confidence intervals are displayed with antenatal healthy AGA ($N$ = 1,216) as the referent group. [a]Low AC velocity, High UT-PI, High UMB-PI, EFW <third centile, Low PAPP-A, sFlt-1:PlGF >38, High AFP; [b]Low AC velocity, High UT-PI, High UMB-PI, EFW <third centile; [c]Low PAPP-A, sFlt-1:PlGF >38, High AFP. Covariates included in all models: maternal factors (age at pregnancy, BMI at recruitment, ethnicity, occupation, partner status, smoking history), infant factors (GA, sex, birth seasonality, childhood physical health), socioeconomic factors (IMD, school funding type, academic year). Abbreviations: AC, abdominal circumference; AFP, alpha-feto protein; AGA, appropriate-for-gestational age; aOR, adjusted odds ratio; CI, confidence interval; EFW, estimated fetal weight; FGR, fetal growth restriction; NA, not applicable; PAPP-A, pregnancy-associated plasma protein-A; sFlt1:PlGF, soluble fms-like tyrosine kinase 1:placental growth factor ratio; SGA, small-for-gestational age; UMB-PI, umbilical artery pulsatility index; UT-PI, uterine artery pulsatility index. Table E. Association between educational attainment aged 7 (Mathematics domain) and markers of placental dysfunction by fetal growth status. Outcome: Not achieving expected educational standard aged 7 in Mathematics domain. Odds ratios (OR) with 95% confidence intervals are displayed with antenatal healthy AGA ($N$ = 1,216) as the referent group. [a]Low AC velocity, High UT-PI, High UMB-PI, EFW <third centile, Low PAPP-A, sFlt-1:PlGF >38, High AFP; [b]Low AC velocity, High UT-PI, High UMB-PI, EFW <third centile; [c]Low PAPP-A, sFlt-1:PlGF >38, High AFP. Covariates included in all models: maternal factors (age at pregnancy, BMI at recruitment, ethnicity, occupation, partner status, smoking history), infant factors (GA, sex, birth seasonality, childhood physical health), socioeconomic factors (IMD, school funding type, academic year). Abbreviations: AC, abdominal circumference; AFP, alpha-feto protein; AGA, appropriate-for-gestational age; aOR, adjusted odds ratio; CI, confidence interval; EFW, estimated fetal weight; FGR, fetal growth restriction; NA, not applicable; PAPP-A, pregnancy-associated plasma protein-A; sFlt1:PlGF, soluble fms-like tyrosine kinase 1:placental growth factor ratio; SGA, small-for-gestational age; UMB-PI, umbilical artery pulsatility index; UT-PI, uterine artery pulsatility index. Table F. Association between educational attainment aged 7 (Science domain) and markers of placental dysfunction by fetal growth status. Outcome: Not achieving expected educational standard aged 7 in Science domain. Odds ratios (OR) with 95% confidence intervals

are displayed with antenatal healthy AGA (N = 1,216) as the referent group. [a]Low AC velocity, High UT-PI, High UMB-PI, EFW <third centile, Low PAPP-A, sFlt-1:PlGF >38, High AFP; [b]Low AC velocity, High UT-PI, High UMB-PI, EFW <third centile; [c]Low PAPP-A, sFlt-1: PlGF >38, High AFP. Covariates included in all models: maternal factors (age at pregnancy, BMI at recruitment, ethnicity, occupation, partner status, smoking history), infant factors (GA, sex, birth seasonality, childhood physical health), socioeconomic factors (IMD, school funding type, academic year). Abbreviations: AC, abdominal circumference; AFP, alpha-feto protein; AGA, appropriate-for-gestational age; aOR, adjusted odds ratio; CI, confidence interval; EFW, estimated fetal weight; FGR, fetal growth restriction; NA, not applicable; PAPP-A, pregnancy-associated plasma protein-A; sFlt1:PlGF, soluble fms-like tyrosine kinase 1:placental growth factor ratio; SGA, small-for-gestational age; UMB-PI, umbilical artery pulsatility index; UT-PI, uterine artery pulsatility index.
(DOCX)

**S6 Table. Association between FGR and educational attainment aged 5–7, stratified based on actual birth weight.** Outcome: Not achieving expected educational standard at each corresponding age. Adjusted odds ratios (OR) with 95% confidence intervals of FGR are displayed with healthy AGA as the referent group. *P* values are based on on logistic regression models of educational performance between 4 antenatal exposure groups: (1) Antenatal FGR; (2) Antenatal healthy SGA; (3) Antenatal AGA with markers of placental dysfunction; and (4) Antenatal healthy AGA. Models are stratified into infants who were SGA vs. AGA at birth. All models are adjusted for the following: maternal factors (age at pregnancy, BMI at recruitment, ethnicity, occupation, partner status, smoking history), infant factors (gestational age, sex, birth seasonality, childhood physical health), socioeconomic factors (IMD, school funding, academic year). Markers of placental dysfunction are defined as one or more of the following: low AC growth between 20–36 weeks, high UT-PI at 20 weeks, high UMB-PI at 36 weeks, EFW <third centile, low PAPPA, sflt1:PlGF ratio, and high AFP. Abbreviations: AC, abdominal circumference; AFP, alpha-feto protein; AGA, appropriate-for-gestational age; aOR, adjusted odds ratio; CI, confidence interval; EFW, estimated fetal weight; FGR, fetal growth restriction; PAPP-A, pregnancy-associated plasma protein-A; sFlt1:PlGF, soluble fms-like tyrosine kinase 1:placental growth factor ratio; SGA, small-for-gestational age; UMB-PI, umbilical artery pulsatility index; UT-PI, uterine artery pulsatility index.
(DOCX)

**S7 Table. Sensitivity analysis comparing EFW threshold <10th vs. <20th percentile when associating antenatal late FGR and educational attainment aged 5–7.** Outcome: Not achieving expected educational standard at each corresponding age. Adjusted odds ratios (OR) with 95% confidence intervals of FGR are displayed with healthy AGA as the referent group. *P values < 0.05, based on logistic regression models of educational performance between 4 antenatal exposure groups: (1) Antenatal FGR; (2) Antenatal healthy SGA; (3) Antenatal AGA with markers of placental dysfunction; and (4) Antenatal healthy AGA. All models are adjusted for the following: maternal factors (age at pregnancy, BMI at recruitment, ethnicity, occupation, partner status, smoking history), infant factors (gestational age, sex, birth seasonality, childhood physical health), socioeconomic factors (IMD, school funding, academic year). Markers of placental dysfunction are defined as one or more of the following: low AC growth between 20–36 weeks, high uterine artery pulsatility index at 20 weeks, high umbilical artery pulsatility index at 36 weeks, EFW <third centile, low PAPPA, sflt1:PlGF ratio, and high AFP.
(DOCX)

**S8 Table. Sensitivity analysis comparing full cohort vs. cohort without subjects with clinically indicated scans at >35 weeks of gestation when associating antenatal late FGR and educational attainment aged 5–7.** Outcome: Not achieving expected educational standard at each corresponding age. Adjusted odds ratios (OR) with 95% confidence intervals of FGR are displayed with healthy AGA as the referent group. *P values < 0.05, based on logistic regression models of educational performance between 4 antenatal exposure groups: (1) Antenatal FGR; (2) Antenatal healthy SGA; (3) Antenatal AGA with markers of placental dysfunction; and (4) Antenatal healthy AGA. All models are adjusted for the following: maternal factors (age at pregnancy, BMI at recruitment, ethnicity, occupation, partner status, smoking history), infant factors (gestational age, sex, birth seasonality, childhood physical health), socioeconomic factors (IMD, school funding, academic year). Markers of placental dysfunction are defined as one or more of the following: low AC growth between 20–36 weeks, high uterine artery pulsatility index at 20 weeks, high umbilical artery pulsatility index at 36 weeks, EFW <third centile, low PAPPA, sflt1:PlGF ratio, and high AFP.
(DOCX)

## Acknowledgments

GS acknowledges the support of GE (provided two ultrasound systems for the POP study) and Roche Diagnostics Ltd (provided equipment and consumable for the analysis of PAPP-A, AFP, and sFlt1:PLGF).

This work was produced using statistical data from ONS. The use of the ONS statistical data in this work does not imply the endorsement of the ONS in relation to the interpretation or analysis of the statistical data. This work uses research datasets which may not exactly reproduce National Statistics aggregates. The views expressed are those of the authors and not necessarily those of the NIHR or the Department of Health and Social Care.

## Author Contributions

**Conceptualization:** Gordon Smith, Catherine Aiken.

**Data curation:** Laurentya Olga, Ulla Sovio, Hilary Wong, Gordon Smith, Catherine Aiken.

**Formal analysis:** Laurentya Olga, Ulla Sovio, Hilary Wong, Catherine Aiken.

**Funding acquisition:** Gordon Smith, Catherine Aiken.

**Investigation:** Ulla Sovio, Gordon Smith, Catherine Aiken.

**Methodology:** Gordon Smith.

**Project administration:** Gordon Smith, Catherine Aiken.

**Supervision:** Gordon Smith, Catherine Aiken.

**Writing – original draft:** Laurentya Olga, Gordon Smith, Catherine Aiken.

**Writing – review & editing:** Ulla Sovio, Hilary Wong, Gordon Smith, Catherine Aiken.

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
