## [Editor Report · Decision Letter 0]

16 Nov 2022

Dear Dr Aiken, 

Thank you for submitting your manuscript entitled "Association between antenatal diagnosis of late fetal growth restriction and educational outcomes in mid-childhood" for consideration by PLOS Medicine.

Your manuscript has now been evaluated by the PLOS Medicine editorial staff as well as by an academic editor with relevant expertise and I am writing to let you know that we would like to send your submission out for external peer review.

Due to current unforeseen staff shortages at PLOS Medicine, editorial decisions are likely to be delayed until the journal is operating at full capacity. While the editorial team will endeavour to ensure a timely handling process, we appreciate that authors may wish to consider submitting manuscripts with particularly time-sensitive findings to an alternative venue at this time. 

Please re-submit your manuscript within two working days, i.e. by Nov 18 2022 11:59PM.

Kind regards,

Callam Davidson

Associate Editor

PLOS Medicine

---

## [Decision Letter · Decision Letter 1]

2 Feb 2023

Dear Dr. Aiken,

Thank you very much for submitting your manuscript "Association between antenatal diagnosis of late fetal growth restriction and educational outcomes in mid-childhood" (PMEDICINE-D-22-03635R1) for consideration at PLOS Medicine. 

[LINK]

In light of these reviews, I am afraid that we will not be able to accept the manuscript for publication in the journal in its current form, but we would like to consider a revised version that addresses the reviewers' and editors' comments. Obviously we cannot make any decision about publication until we have seen the revised manuscript and your response, and we plan to seek re-review by one or more of the reviewers. 

We hope to receive your revised manuscript by Feb 23 2023 11:59PM. Please email us (plosmedicine@plos.org) if you have any questions or concerns.

We look forward to receiving your revised manuscript. 

Sincerely,

Callam Davidson, 

PLOS Medicine

plosmedicine.org

Please revise your title according to PLOS Medicine's style. Please place the study design ("A cohort study”) in the subtitle (ie, after a colon).

In the last sentence of the Abstract Methods and Findings section, please describe the main limitation(s) of the study's methodology.

The Data Availability Statement (DAS) requires revision. For each data source used in your study: 

a) If the data are owned by a third party but freely available upon request, please note this and state the owner of the data set and contact information for data requests (web or email address). Note that a study author cannot be the contact person for the data.

b) If the data are not freely available, please describe briefly the ethical, legal, or contractual restriction that prevents you from sharing it. Please also include an appropriate contact (web or email address) for inquiries (again, this cannot be a study author).

Thank you for including an Author Summary. The third bullet point under ‘Why was this study done’ appears better suited to ‘What did the researchers do and find?’.

Please describe and cite your Supporting Information items as outlined here: https://journals.plos.org/plosmedicine/s/supporting-information#loc-item-description

Line 209: For consistency, please place numerators/denominators within and percentages outside of parentheses. 

Please remove the Competing interests, Funding, and Author contributions from the main text – all are captured as metadata via the submission form questionnaire. The acknowledgements can be kept in the main text. 

Please add to the competing interests (submission form): GS serves on PLOS Medicine’s editorial board. 

For [Internet] sources in the References, please include the date accessed. 

Comments from the reviewers:

Reviewer #1: Alex McConnachie, Statistical Review

This review considers the statistical aspects of the paper by Olga et al, looking at fetal growth restriction and measures of educational attainment at age 5-7.

The basic approach of using logistic regression, with adjustment for potential confounders, to estimate associations between the exposures of interest and failure to achieve the expected educational milestones is reasonable, though by dichotomized the outcomes, there is a risk that the study is losing power. It could have been more efficient to look at the outcomes as continuous measures, though this could be difficult, depending on the distribution of the outcomes.

The main issue with the analysis is that the authors have taken two separate exposures (small for gestational age, and placental dysfunction) and combined these to produce a single variable with 4 levels. They then look at each of the three subgroups with one or both exposures, and compare them to the subgroup with neither exposure. They interpret a "significant" p-value as giving evidence of an association, and a "non-significant" p-value as meaning that the outcomes were comparable. This is a false argument.

For example, looking at supplementary table 4B, at age 7, the adjusted OR for reading for FGR vs Healthy AGA is 1.46, p=0.05, whereas for Health SGA vs Healthy AGA, the adj OR is 1.41, p=0.2. Here, the ORs are quite similar, but only for the FGR subgroup is it statistically "significant", since the sample size is bigger (250 FGR, 126 Healthy SGA). The adj OR for AGA with placental dysfunction is 0.93, which is quite close to 1, so for this outcome the data appear consistent with there being an association with SGA, but not placental dysfunction.

On the other hand, for writing at age 7, the adj ORs are 1.46, 0.98, and 1.25 for FGR, AGA with placental dysfunction, and Healthy SGA, respectively. This values are consistent with there being an association with SGA, which is amplified in the presence of placental dysfunction. Without SGA, placental dysfunction has no association with achievement, but in the presence of SGA, placental dysfunction makes achievement in writing at age 7 worse.

A better modelling approach might have been to first assess whether SGA and/or placental dysfunction were independently associated with educational outcomes, and then to test whether there is an interaction. If the FGR group have the lowest educational achievement, this could simply be because they have both risk factors, which independently predict poorer performance. Alternatively, it could be that the FGR group does particularly poorly, which would be evidenced by an interaction between the two exposures.

Unfortunately, there may be little power to detect interactions with this sample size. Nevertheless, the authors need to avoid simply interpreting their results in terms of whether the p-values are less than 0.05. At least as much emphasis should be put on the sizes of the associations when interpreting the results.

Minor points:

The STROBE checklist is used, but should REPORT extension have been used, since routine data linkage has been performed?

Line 177 states that "…childhood morbidities not linked to intrauterine development could potentially confound the relationship between perinatal factors and mid-childhood educational attainment." I am probably misunderstand something, but if they are not linked to IU development, how can they be confounders? Besides, why does it matter if the association between perinatal factors and educational achievement is confounded, if that is not the focus of the analysis?

Reviewer #2: PLOS One 

Editor in Chief

December 4, 2022

Re: Laurentya Olga et al. Association between antenatal diagnosis of late fetal growth restriction and educational outcomes in mid-childhood

Comments for the authors:

The study entitled 'Association between antenatal diagnosis of late fetal growth restriction and educational outcomes in mid-childhood' focuses on reading, writing and mathematical skills of children born with fetal growth restriction (FGR). From a clinical point of view, the study is very relevant. FGR studies with long-term outcomes are needed to optimize the timing of the delivery, to counsel the parents and to develop preventive strategies in order to help these children to reach their full academic potential. 

In this cohort study, antenatal assessments included ultrasound examination (estimated fetal weight (EFW), uterine and umbilical artery Doppler measurements) as well as maternal serum concentrations of PAPP-A, AFP, s-Flt and PlGF at 20, 28 and 36 gestational weeks. The study cohort was divided to four antenatal phenotype groups for comparisons of educational outcomes. My comments concerning the manuscript are listed below, the major ones being numbers 2 and 4. I understood that a statistician will separately review the manuscript, and therefore this review does not include any comments on the applied statistics. 

1. According to CDC (Centers for Disease Control and Prevention) middle childhood is determined as age 6-8 years. Is it appropriate to use term mid-childhood in the title as the study results range from 5-7 years?

2. The researchers divided the cohort to four groups antenatally 1) FGR, 2) healthy SGA, 3) appropriate for gestational age (AGA) with markers of placental dysfunction and 4) healthy AGA (reference group). FGR was defined as EFW < 10th centile and any of the following: (a) lowest decile of abdominal circumference (AC) growth velocity between 20 and 36 weeks, (b) EFW <3rd centile at 36 weeks, c) umbilical artery pulsatility index in the highest decile at 36 weeks, d) uterine artery pulsatility index in the highest decile at 20 weeks, e) maternal PAPP-A <0.4 MoM at 12 weeks, f) maternal sFlt1:PlGF- ratio >38 at 36 weeks, g) maternal AFP >2.0 MoM at 20 weeks. The researchers explain that they have tried to phenotype the groups carefully in order to distinguish constitutionally small children from children diagnosed antenatally with FGR/placental insufficiency. 

My main concern regarding this study is that this phenotyping is not optimal for late FGR. Several FGR inclusion criteria in this study are used to predict the development of FGR/indicate an increased FGR risk, but they are not indicative of FGR per se. The researchers could phenotype FGR in this large cohort more precisely according to internationally accepted criteria for FGR. In addition, longitudinal development of placental insufficiency, for example umbilical artery PI findings at earlier time points would ensure the readers that only late FGR cases were included. Naturally, it would be also interesting to know the results in various subgroups, formed according to the measured biomarkers, in this exceptional data set. 

3. The growth charts used in this study are from 1990. Any newer charts available?

4. In healthy AGA group, 21% of the children performed below the standard in overall educational attainment at 5 years of age and 23% performed below the standard in writing skills at 7 years of age. What is the reason that every fourth - fifth AGA child with no signs of placental insufficiency did not to achieve the standard in these educational tests? The educational standard, other health/educational issues in the AGA group despite extensive confounding factor evaluation, teaching, or what? We should be worried about this proportion; one can even argue that this is a greater problem than 5% additional increase in poor performance in writing skills at 7 years in the FGR group compared to this healthy AGA group. 

5. Could a more detailed discussion on possible mechanisms behind the educational problems in FGR included? The authors discuss that early iatrogenic delivery might have an input. On the other hand, one can argue that delaying the delivery might result in non-optimal outcome. 

6. There are publications focusing on reading, writing and other educational outcomes in FGR children with placental insufficiency for example by Geva et al and Partanen et al, which would enhance the findings of the current study and could be discussed. 

Reviewer #3: The objective of this study was to determine if fetal growth restriction diagnosed in the prenatal period is associated with poorer education in childhood. 

The study is based on the pregnancy outcomes prediction study that included unselected nulliparous women between 2008 and 2012. The samples included 4200 patients. Three visits were conducted for research purpose at 20, 28, 36 weeks. Several analytes were measured in peripheral blood including PAPP-A, AFP, PlGF and s-Flt. This study is unique in many ways and could lead to important publication. The outcomes of children in term of education were obtained from the national database in England. Standardized assessments are available for the age of 5, 6, and 7. The article follows the STROKE guidelines which are in Appendix 3. To avoid the confounding effect of preterm birth, the analysis was restricted to infants born at 36 weeks and beyond. 

Imputation for covariant was performed using chained equation under a missing at random assumption (detail on page 9). 

Results: 

* 65% of patients were included in the analysis 

* Mean gestational age at delivery 40 weeks

* The prevalence of fetal growth restriction was 9%

* Education at ages 5 and 6: antenatal diagnosis of FGR had a higher rate of not meeting educational standards compared to AGA (lines 220-224). 

* Education at age 7: overall the proportion failing to attend expected education were 16% for reading, 22% for writing, 17% for mathematics and 10% for science. However, the risk was higher for FGR although OR were 1.46-1.49 for the different domains. 

Assessment: I find the article very well written and is straightforward. The results are clear, and the conclusions are justified by the data. The results have a lot of information for patients counselling and management. 

Questions for the authors:

Fetal growth restriction was defined as a combination of biometric parameters and one or more markers of placental dysfunction. Most obstetrical units use only biometry as an estimated fetal weight less than 10th percentile. What would be the results if the authors use only fetal biometry such as estimated fetal weight below 10th centile, change in abdominal circumference, etc.? I wonder if babies who are SGA without evidence of growth restriction would also have normal education retainment. It would be worth for the obstetrical community to have this information because many around the world do not use PAPP-A or the sFlt-1/PlGF ratio.

[LINK]

---

## [Decision Letter · Decision Letter 2]

21 Mar 2023

Dear Dr. Aiken,

Thank you very much for re-submitting your manuscript "Association between antenatal diagnosis of late fetal growth restriction and educational outcomes in mid-childhood: A prospective cohort study with long-term data linkage" (PMEDICINE-D-22-03635R2) for review by PLOS Medicine.

I have discussed the paper with my colleagues and the academic editor and it was also seen again by two reviewers. I am pleased to say that provided the remaining editorial and production issues are dealt with we are planning to accept the paper for publication in the journal.

The remaining issues that need to be addressed are listed at the end of this email. Please take these into account before resubmitting your manuscript:

We look forward to receiving the revised manuscript by Mar 28 2023 11:59PM.   

Sincerely,

Callam Davidson,

Associate Editor 

PLOS Medicine

plosmedicine.org

Requests from Editors:

Please include the study setting in the title.

The Data Availability Statement (DAS) requires revision. For the raw data from the POPS study, please include an appropriate contact (web or email address) for inquiries (this cannot be a study author).

Please quantify the key findings from the study in your Author Summary. 

Thanks for including a completed RECORD checklist - when completing the checklist, please use section and paragraph numbers, rather than page numbers (as these will change during the publication process).

Please remove the Data Availability Statement from the main text and ensure all the relevant information is captured in the submission form questionnaire (your response will be published alongside the article).

Changes from the planned analysis should be identified as such in the Methods section of the paper, with rationale.

Comments from Reviewers:

Reviewer #1: Alex McConnachie, Statistical Review

I thank the authors for their consideration of my original points. I am generally happy with their responses.

I note that a power calculation has been added, but this is not reproducible as it stands. It would help if the authors give the assumed percentages in the FGR and healthy AGA groups. If the calculation was done prior to the study, then the numbers required in each group to achieve 80% power should be given. If this calculation was post-hoc, based on the numbers achieved, then this should be stated.

For the healthy SGA vs. healthy AGA comparisons, it is OK to say that that the CIs are wide, but given the newly added power calculation, it would seem that this comparison is very underpowered. Someone only reading the abstract, lines 44-45, or lines 69-70 in particular, will not get that nuance, and will infer that the only subgroup at increased risk of poor achievement is the FGR group. When making headline statements, the authors could be a little more careful.

As an afterthought, I wonder whether there is any possibility that some of those in the healthy SGA group could in fact be undiagnosed FGR - could there be a subgroup of false negatives? What about false positives? How good are the tests for FGR? How would misdiagnoses affect the results? This could, perhaps, be discussed in the limitations section.

---

## [Editor Report · Decision Letter 3]

28 Mar 2023

Dear Dr Aiken, 

On behalf of my colleagues and the Academic Editor, Professor Jenny Myers, I am pleased to inform you that we have agreed to publish your manuscript "Association between antenatal diagnosis of late fetal growth restriction and educational outcomes in mid-childhood: A UK prospective cohort study with long-term data-linkage study" (PMEDICINE-D-22-03635R3) in PLOS Medicine.

When making the formatting changes, please also address the following editorial comment:

* Lines 200-203: 'Sensitivity analyses by using EFW threshold <20th percentile or excluding

201 women with ultrasound scan >35 weeks of gestational age did not substantively alter the results and

202 so these have not been reported here'. Please include the sensitivity analyses as Supporting Information and cite them here.

PRESS

Sincerely, 

Callam Davidson 

Associate Editor 

PLOS Medicine